# Diagnostic Value of the Alveolar–Arterial Oxygen Gradient in Pulmonary Embolism: A Cross-Sectional Study

**DOI:** 10.3390/healthcare13010011

**Published:** 2024-12-24

**Authors:** Ana Maslac, Slavica Juric Petricevic, Miro Vukovic, Ivan Skopljanac

**Affiliations:** 1School of Medicine, University of Split, 21000 Split, Croatia; anamaslac1@gmail.com (A.M.); mirovukovi0@gmail.com (M.V.); 2Pulmonology Department, University Hospital of Split, 21000 Split, Croatia; slavicajuric01@gmail.com

**Keywords:** pulmonary embolism, alveolar–arterial gradient, CT angiography of the pulmonary artery, D-dimers

## Abstract

**Background/Objectives**: Pulmonary embolism (PE) is a potentially serious condition characterized by the blockage of blood vessels in the lungs, often presenting significant diagnostic challenges due to its non-specific symptoms. This study aimed to evaluate the utility of the alveolar–arterial (A-a) oxygen gradient as a diagnostic tool for PE, hypothesizing that it could enhance early detection when combined with other clinical markers. **Methods**: We retrospectively analyzed 168 patients at the University Hospital Center Split. This study correlated A-a gradients with PE confirmed by CT pulmonary angiography. Key clinical and biochemical markers, including heart rate, C-reactive protein (CRP), pro-brain natriuretic peptide (NT-proBNP), D-dimer, high-sensitivity troponin (hs-troponin), and arterial oxygen pressure (PaO_2_), were assessed. **Results**: Our findings revealed that patients with PE had significantly higher A-a gradients than those without PE. The observed-to-expected ratio for the A-a gradient was notably increased in the PE group. Additionally, patients with PE exhibited elevated heart rate, CRP, NT-proBNP, D-dimer, and hs-troponin levels, while PaO_2_ levels were notably lower. **Conclusions**: This study demonstrates that an elevated A-a gradient reflects the severity of gas exchange impairment in PE. The results suggest that early diagnosis of PE may be improved by incorporating A-a gradient analysis alongside other clinical markers, potentially leading to more effective and timely interventions.

## 1. Introduction

Venous thromboembolism (VTE), which includes deep vein thrombosis (DVT) and pulmonary embolism (PE), is the third most common cardiovascular disease [1]. A PE occurs when a thrombus blocks the pulmonary vessels, typically from the pelvis or lower extremities. There are also other causes, including air embolism, amniotic fluid, or fat embolism [2]. A European incidence of 50 cases per 100,000 has been reported [3], though PE can also be asymptomatic or lead to sudden death, complicating understanding of its epidemiology. Over 317,000 deaths occurred in the EU in 2004 due to it, making it one of the most common causes of death and morbidity [4]. There are several risk factors for this condition, including surgery, trauma, and immobilization [5]. In severe PE, right ventricular failure caused by increased afterload is the leading cause of death [6].

PE diagnosis can be missed due to its nonspecific symptoms, such as chest pain and shortness of breath [7,8]. Although the Wells criteria are widely used for assessing PE probability, other scores such as the Geneva score and the YEARS criteria provide objective and simplified approaches for estimating PE likelihood [9]. D-dimers, with their high negative predictive value, remain critical in excluding PE, particularly in low- and intermediate-risk patients. hs-Troponin levels, indicative of myocardial strain, can provide additional insights into right ventricular failure associated with PE [10]. The gold standard for diagnosis remains CT pulmonary angiography [11].

The A-a oxygen gradient is an important indicator of pulmonary function, particularly in assessing gas exchange efficiency between the alveoli and arterial blood. The A-a gradient quantifies the disparity between the oxygen partial pressure in the alveoli (PAO_2_) and that in arterial blood (PaO_2_) [12], thereby aiding in the identification of the etiology of hypoxemia. A typical A-a gradient implies that hypoxemia may stem from factors such as hypoventilation, whereas an increased gradient signifies disrupted oxygen diffusion, which is characteristic of conditions such as pneumonia or pulmonary embolism [13,14].

Consequently, the A-a gradient proves to be an important tool in differentiating oxygenation-related problems. In PE, the A-a gradient is generally high due to ventilation–perfusion mismatch in the presence of obstruction to the pulmonary arterial tree leading to ventilation of areas not being perfused. The end result is poor arterial oxygenation with a normal or near-normal alveolar oxygen level [15]. By quantifying impairment in oxygenation, the A-a gradient could be applied to support findings for the order of follow-up diagnostic imaging such as CT pulmonary angiography. The A-a gradient could particularly be valuable for identifying low-probability PE patients where CT pulmonary angiography may be avoided. Although CTA is currently the gold standard for the diagnosis of PE, this imaging modality is relatively expensive and not as widely available which would include sending patients to bigger centers. CTA also involves radiation exposure, which is especially concerning for pregnant women, in whom alternative diagnostic pathways are preferred to avoid radiation risks. More specific clinical predictors are required to help justify the use of this imaging technique and to limit unnecessary testing, such as an elevated A-a gradient.

The purpose of this study was to evaluate the diagnostic value of the alveolar–arterial oxygen gradient across different thrombus burden groups in the diagnosis of pulmonary embolism and assess its potential role in guiding clinical decision-making.

## 2. Materials and Methods

This is a cross-sectional study conducted at the Clinic for Pulmonary Diseases and the Clinical Department of Diagnostic and Interventional Radiology at the University Hospital Center Split, during the period from January 2019 to December 2023. Patients were included based on clinical suspicion of PE, referral for CT pulmonary angiography, and completion of arterial blood gas analysis upon admission. The following parameters were used in this study: age and gender of the patients, oxygen saturation, arterial oxygen pressure, arterial carbon dioxide pressure, CT pulmonary angiography findings, troponin levels, CRP, and NT-proBNP. Based on anamnesis data, risk factors were extracted. This study was also in compliance with the provisions of the Code of Medical Ethics and Deontology (NN 55/08, 139/15) and the Helsinki Declaration (1964–2013).

Inclusion criteria were:Patients who underwent CT pulmonary angiography due to suspected pulmonary embolism.Patients who underwent arterial blood gas analysis.

Exclusion criteria were defined to ensure the accuracy and clarity of this study’s findings, minimizing potential confounding factors and uncertainties:Patients with lung conditions, such as lung tumors exceeding 2 cm in size or pulmonary edema.Patients whose CT readings did not clearly define the presence or absence of pulmonary embolism and instead stated “pulmonary embolism diagnosis is possible”.

To ensure accuracy, ABG measurements included only those performed before initiating oxygen therapy, which is standard protocol upon emergency admission. If oxygen was administered prior, the analysis was excluded to maintain consistency. This study included 168 patients aged 17 to 91 years. The inclusion of patients aged 17 was based on hospital policy allowing emergency admission for younger patients in adult care during specific circumstances. All patients were examined upon emergency admission. The examination included a detailed anamnesis with a focus on risk factors for pulmonary embolism, where it was noted whether the patient had active cancer, hemoptysis related to the current condition, a history of pulmonary embolism or deep vein thrombosis, or any surgical procedures within the last 4 weeks. Each patient underwent a clinical examination, where heart rate was recorded, along with the presence of clinical signs of DVT. Clinical signs of DVT were assessed using the Wells score criteria, which include heart rate, unilateral leg edema, immobilization, and haemoptysis. Blood was drawn for hematology, coagulation, and biochemistry tests. An arterial blood sample was taken for gas analysis, which is simple, painless, and provides immediate results. The ABL90 FLEX device from Radiometer was used for this calculation in the emergency department. The recorded data of interest included oxygen saturation of arterial blood, arterial oxygen pressure, and arterial carbon dioxide pressure. When a working diagnosis of PE was made, the patient was sent for CT pulmonary angiography with Optiray contrast.

The A-a gradient was calculated using the following equation:A-a O_2_ gradient = PAO_2_ − PaO_2_ (measured from ABG)

PAO_2_ was derived from the formula:PAO_2_ = FiO_2_ × (Atmospheric Pressure − H_2_O Pressure) − (PaCO_2_/Respiratory Quotient)

FiO_2_ was set to 21% for all patients breathing room air.

Atmospheric pressure was assumed to be 760 mmHg at sea level, and H_2_O pressure was standardized to 47 mmHg.

The respiratory quotient (R) was assumed to be 0.8, which is standard in clinical practice.

This calculation provided individualized A-a gradients based on the patient’s arterial blood gas values, allowing us to identify deviations indicative of pulmonary embolism.

As such, it is not possible to use a single cut-off value for all patients, and the gradient must be interpreted in the context of the patient’s age.

The validation set consisted of patients who had normal CT angiography results from 2019 to 2023. Patients who did not have a pulmonary embolism were not followed up to get a final diagnosis as these analyses were made on the emergency department documentation, which did not delve into other diagnoses as soon as PE was excluded by CT. This study focused on the diagnostic significance of the A-a gradient in diagnosing pulmonary embolism, hence additional examination of non-PE diagnoses was outside the scope of this study.

All statistical calculations and analyses were performed using SPSS (Statistical Package for the Social Sciences) version 26.0 and R version 4.0.5. To assess differences between groups (patients with and without PE), the Mann–Whitney U test was used, as the data on the A-a ratio values showed an asymmetric distribution. The Mann–Whitney U test is a non-parametric test that compares the medians of two independent groups. The significance of differences between groups was assessed at a significance level of *p* < 0.05. For analyzing the distribution of binary outcomes in this study, we used the chi-square test, which assesses whether there is a statistically significant difference in frequency between two or more categorical variables. To evaluate the diagnostic accuracy of the ratio of expected and observed A-a values in predicting PE, a ROC curve (Receiver Operating Characteristic) analysis was used, and the AUC (Area Under the Curve) value was calculated. The ROC curve shows the ratio of sensitivity and 1-specificity across different threshold values (cut-off), and the AUC value serves as an overall measure of the model’s diagnostic accuracy. To determine the optimal cut-off for the ratio of expected and observed A-a values, the Youden index (J) was used, which maximizes the combination of sensitivity and specificity. We also analyzed the A-a gradient across thrombus burden groups, including no embolism, massive embolism, lobar embolism, segmental embolism, and subsegmental embolism. Thrombus burden was categorized based on radiological findings assessed by a single radiologist. Median A-a gradient values and interquartile ranges were calculated for each group, and differences were evaluated using the Kruskal–Wallis test. Post-hoc Dunn analysis was performed to identify significant differences between groups. The statistical significance threshold was set at *p* < 0.05.

## 3. Results

In this study, patients with pulmonary embolism exhibited significantly elevated A-a gradient values compared to those without embolism. The median A-a gradient values were higher across all thrombus burden groups (massive, lobar, segmental, subsegmental) when compared to the no-embolism group (*p* < 0.006). However, no significant differences were observed among the embolism groups themselves.

### 3.1. Initial Values of Parameters in Participants

A total of 168 patients from the University Hospital Center Split who underwent CT pulmonary artery angiography due to suspected pulmonary embolism were included in this study. The median age was 69, and 53% of the patients were female. Seven patients had pneumonia without PE, but due to the inflammation, their A-a gradient was increased, so they were excluded from this study. The general characteristics of the participants, including their laboratory parameters and CT scan results, are shown in Table 1. Thirty-three participants with confirmed PE had normal PaO_2_ (35%), of which 6 (7%) had a normal A-a gradient. Out of 93 patients with pulmonary embolism, 15 (16%) had a pleural effusion, while 8 (9%) exhibited pleural effusion combined with pulmonary parenchymal consolidation. Ground-glass opacities were detected in 7 (8%) patients, tumors in 4 (4%) patients, bronchiolitis in 1 (1%) patient, and isolated atelectasis in 1 (1%) patient. Additionally, 1 patient had both atelectasis and pulmonary infarction. Of all the patients with additional findings on CT pulmonary artery angiography, only two patients, who had both PE and pleural effusion, exhibited a reduced A-a gradient.

When comparing the groups of patients with confirmed and unconfirmed PE, heart rate, D-dimer, hs-Troponin, CRP, and NT-proBNP levels were significantly higher in patients with confirmed PE, while PaO_2_ values were lower in patients with confirmed PE (Mann–Whitney test, Table 2).

### 3.2. Comparison of A-a Gradient in Participants with Confirmed and Unconfirmed PE

This study analyzed the absolute frequencies of pulmonary artery CT angiography outcomes in relation to A-a values. It was shown that among the participants for whom CT did not confirm PE, 29 had a normal A-a gradient and 45 had an elevated A-a gradient. Seven participants for whom CT confirmed PE had a normal A-a gradient, and 86 had an elevated A-a gradient. Statistical analysis showed a significant difference between the groups with normal and elevated A-a gradients (*p* < 0.001 according to the chi-square test), indicating a significant correlation between the gradient and CT scan outcomes in the diagnosis of pulmonary embolism, as shown in Figure 1.

For further analysis, we calculated the ratio of expected and observed A-a values to quantify how much each patient differed from the expected value. Using a continuous outcome (ratio of expected and observed A-a values) instead of a binary one (normal vs. abnormal A-a value) allows for broader statistical analysis. After separating the total sample of participants based on the presence or absence of PE, it was shown that the group with PE had a significantly higher median ratio of expected and observed A-a values (2.14 vs. 1.18, *p* < 0.001, Mann–Whitney test). Corresponding median values with a 95% confidence interval are shown in Figure 2.

When we applied our data to logistic regression models with the ratio of expected and observed A-a values as the independent variable, the analysis resulted in a model with an AUC value of 0.824 (*p* < 0.001) and a Nagelkerke R^2^ value of 0.3926 (Figure 3).

The calculated odds ratio with the Youden index is shown in Table 3. If the ratio of the expected and observed A-a gradient were used in diagnosing PE, the results of this model suggest that a reference value of 1.483 would result in 75% sensitivity and 78% specificity of such a test.

The A-a gradient was analyzed across the thrombus burden groups: no embolism, massive, lobar, segmental, and subsegmental embolism (Figure 4). Median A-a gradient values and interquartile ranges for each thrombus burden group were as follows: no embolism (median 2.919, IQR 1.850–3.612), massive embolism (median 5.882, IQR 4.561–8.086), lobar embolism (median 5.244, IQR 3.600–7.032), segmental embolism (median 5.887, IQR 4.409–7.214), and subsegmental embolism (median 3.790, IQR 3.550–6.138). While variations in median A-a gradient values were observed among the massive, lobar, segmental, and subsegmental embolism groups, statistical analysis revealed no significant differences between these categories of PE. Post-hoc Dunn analysis confirmed that the A-a gradient values in the massive, lobar, segmental, and subsegmental embolism groups were significantly higher than in the no-embolism group (*p* < 0.000001). 

## 4. Discussion

Objective diagnosis of pulmonary embolism is crucial, as clinical assessment alone is unreliable, and the consequences of misdiagnosis are severe. Delayed diagnosis of pulmonary embolism is associated with a high mortality rate, while a false positive diagnosis can unnecessarily expose patients to the risks of anticoagulant therapy [16]. In this study, we analyzed the correlation between the expected and observed A-a gradient ratio and the occurrence of pulmonary embolism. The results show that the A-a ratio was significantly higher in patients with PE (median 2.14) compared to those without PE (median 1.18), with a *p*-value of less than 0.001. ROC analysis indicated that this model could effectively distinguish between patients with and without pulmonary embolism, with an AUC value of 0.824. The comparison of groups with confirmed and unconfirmed PE revealed significant differences in the analyzed parameters, including heart rate, CRP, NT-proBNP, D-dimers, hs-troponin, and PaO_2_. The *p*-value for CRP, D-dimers, hs-troponin, and PaO_2_ was extremely low, *p* < 0.001, indicating a very high statistical significance of the results. These findings suggest that the parameters studied are useful in distinguishing between patients with and without PE, which can play an important role in diagnosing and monitoring this clinical condition. Murill et al. reported in their study of 50 patients that decreased PaO_2_ is one of the indicators of PE [17]. In our study, 35% of patients with PE had normal PaO_2_, indicating that the presence of normal oxygen levels in the blood does not exclude the diagnosis of PE, and only 7% had a normal A-a gradient, which supports the value of the A-a gradient as a diagnostic parameter.

The results obtained show that the ratio of expected and observed A-a gradients can be a useful diagnostic tool for identifying patients with PE. With a sensitivity of 75.27% and a specificity of 78.38%, this ratio can provide important information to aid clinical decision-making, particularly in emergency situations where a rapid diagnosis of PE is required. This method can be used as an additional diagnostic tool in combination with existing methods to improve the accuracy, efficiency, and speed of the diagnostic process. Including the A-a gradient ratio in existing screening algorithms for diagnosing PE can help reduce the need for invasive procedures such as CT angiography, especially in low-risk patients.

Our findings emphasize the limitations of ABG analysis in detecting PE when compared to the A-a gradient. This discovery could imply the importance of the A-a gradient in detecting major anomalies even in patients with normal PaO_2_ levels but further studies in larger populations are needed. Patients with normal A-a gradients but confirmed PE represent a unique subset, possibly explained by compensatory mechanisms maintaining gas exchange. In response to impaired gas exchange caused by V/Q mismatch, patients often hyperventilate. This increases alveolar oxygen compensating for the reduced oxygen delivery in obstructed regions of the pulmonary circulation. While this preserves normal arterial oxygen pressure it simultaneously increases the A-a gradient by decreasing pCO_2_. This implies that the A-a gradient may be a more sensitive marker than ABG of underlying ventilation–perfusion abnormalities.

D-dimers stand out as a parameter in the diagnosis of pulmonary embolism precisely because of their high negative predictive value, especially in low- and intermediate-risk patients, as demonstrated by Gupta et al. in their study [18]. As such, D-dimers could be used in combination with the A-a gradient for a more reliable diagnosis of PE. McFarlane et al. achieved similar results to ours, showing that a normal A-a gradient in patients without a history of DVT or PE makes the diagnosis of PE less likely. Additional diagnostic evaluation may be unnecessary in this subgroup of patients [15].

It would be useful to observe the value of the A-a gradient in relation to the mortality of patients with pulmonary embolism. The relationship between the A-a gradient and the severity of the pulmonary embolism could also be tracked simultaneously, which unfortunately was not within the scope of this study. Our analysis of the radiological severity of PE highlights that all patients with pulmonary embolism, including those with subsegmental and segmental thrombi, exhibited elevated A-a gradients. This finding underscores the sensitivity of the A-a gradient to gas exchange abnormalities, even in cases with minimal thrombus burden.

Due to its great potential, the A-a gradient can be a useful tool in diagnosis, allowing comparison with other methods for detecting pulmonary embolism. Our results are consistent with a retrospective observational study by Karakayal et al., who found a weak but positive correlation between the pulmonary artery obstruction index (PAOI) and the A-a gradient. Their research showed that the values of the alveolar–arterial oxygen gradient can be clinically useful, as such testing is simple to perform and cost-effective. Moreover, it can be performed at the bedside, and results can be obtained in less than a minute [19].

The main limitation of this study is the small population sample, with data collected retrospectively from 2019 in only one clinical center. The alveolar–arterial gradient has proven to be a useful diagnostic tool in evaluating patients whose chest X-ray findings are nonspecific and could not explain the patient’s complaints and symptoms of dyspnea. Additional research should be conducted in a population of patients who had a pulmonary substrate. The A-a gradient is influenced by various factors, including respiratory drive, concurrent lung pathologies (e.g., pneumonia or atelectasis), and patient-specific variables such as age and hemoglobin levels. Additionally, the assumption of a uniform respiratory quotient (0.8) introduces variability. These factors underscore the importance of interpreting the A-a gradient in the context of the clinical status and not as a standalone diagnostic tool. Furthermore, in subsegmental and segmental PE, the A-a gradient may be useful, though its sensitivity may decrease due to a lower thrombus burden. Assessment of thrombus burden was performed by a single radiologist, which introduces the potential for subjectivity in categorizing the severity and radiological assessment may not fully reflect the hemodynamic or clinical impact of the thrombus burden. More studies are needed to establish its role in these subgroups. Another study limitation is the lack of definitive diagnosis for individuals without verified pulmonary embolism. While this information may give important clinical context, the major goal of this study was to determine the usefulness of the A-a gradient in diagnosing PE.

## 5. Conclusions

The current study demonstrates that the A-a gradient ratio could be a useful tool in the diagnosis of pulmonary embolism, especially if used in combination with other diagnostic tools, such as D-dimers and clinical scores such as the Wells criteria and Geneva score. The addition of the A-a gradient to existing diagnostic algorithms may provide a cost-effective, rapid, and noninvasive approach that may improve diagnostic accuracy, especially in emergency situations where rapid assessment is critical. These findings require further confirmation in larger samples and in more diversified clinical settings in establishing the relationship of A-a gradients with mortality in patients with PE.

## Figures and Tables

**Figure 1 healthcare-13-00011-f001:**
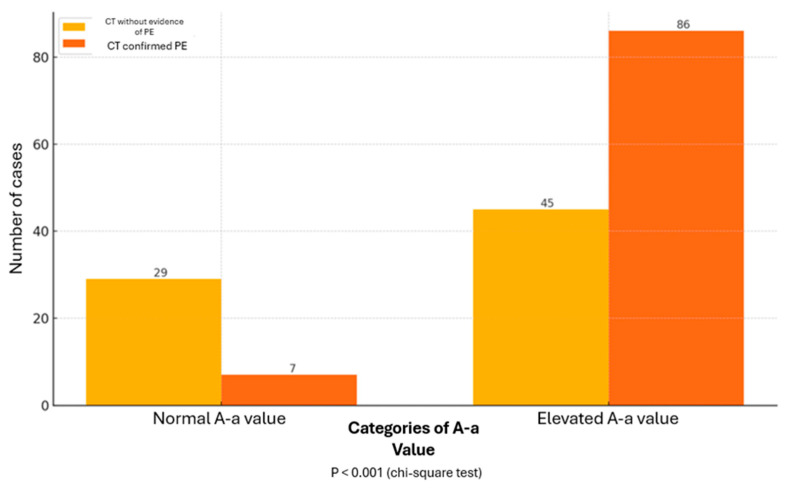
Absolute Frequencies of CT Angiography Outcomes Concerning A-a Values.

**Figure 2 healthcare-13-00011-f002:**
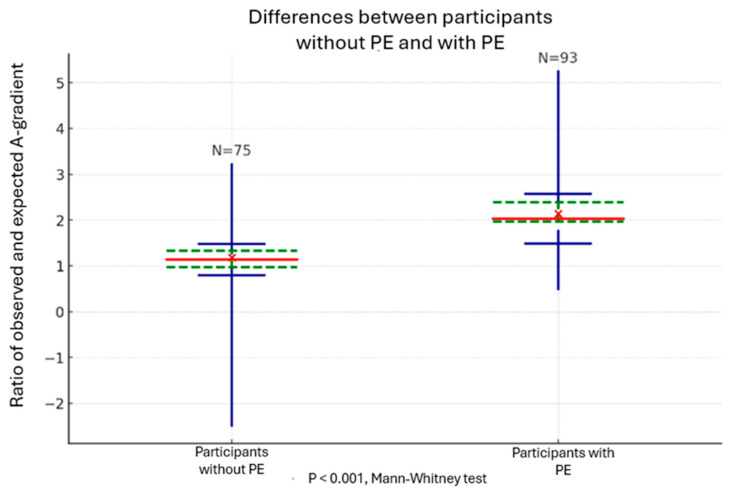
Differences Between Median Values of the Ratio of Expected and Observed A-a Gradients Between Participants Without PE and With PE. Red Lines: median ratio. Green Dashed Lines: interquartile range (IQR). Horizontal Blue Lines: confidence interval (CI).

**Figure 3 healthcare-13-00011-f003:**
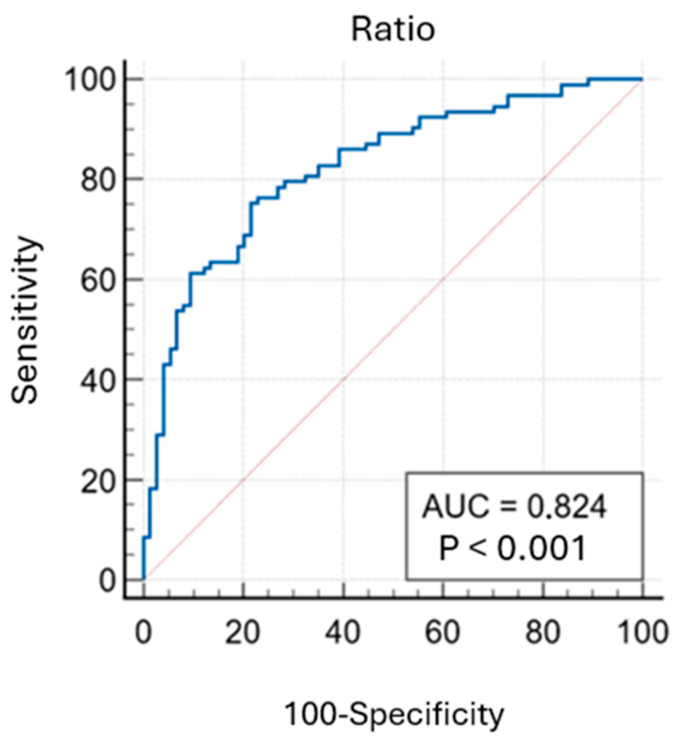
ROC curve for logistic regression model with the ratio of expected and observed A-a gradient (AUC = 0.824, *p* < 0.001, Nagelkerke R^2^ = 0.392).

**Figure 4 healthcare-13-00011-f004:**
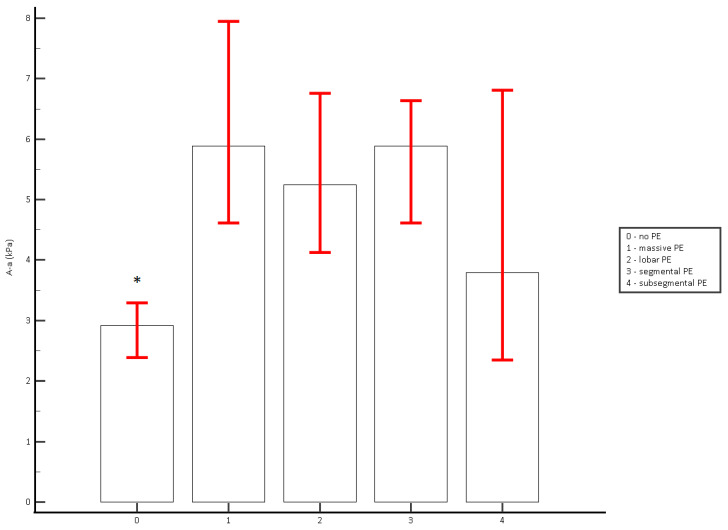
Median A-a gradient by Thrombus Burden Group. * A-a gradient values in the massive, lobar, segmental, and subsegmental embolism groups were significantly higher than in the no-embolism group (*p* < 0.000001). Error bars represent the 95% confidence intervals for the medians.

**Table 1 healthcare-13-00011-t001:** Basic Characteristics of Participants.

Gender [n (%)]	
Male	79 (47)
Female	89 (53)
Age [Median (IQR)]	69 (17–92)
Active cancer [n (%)]	37 (22)
Previous DVT and/or PE [n (%)]	34 (20)
Surgery within 4 weeks [n (%)]	13 (8)
Clinical signs of DVT [n (%)]	20 (12)
Hemoptysis [n (%)]	12 (7)
PE negative [n (%)]	74 (44)
PE positive [n (%)]	93 (56)
Left-sided PE [n (%)]	14 (15)
Right-sided PE [n (%)]	31 (34)
Bilateral PE [n (%)]	47 (51)
Massive PE [n (%)]	16 (17)
Lobar PE [n (%)]	38 (41)
Segmental PE [n (%)]	31 (34)
Subsegmental PE [n (%)]	7 (8)

Data are presented as: n (%) or median (IQR).

**Table 2 healthcare-13-00011-t002:** Biochemical and Clinical Parameters between Positive and Negative Pulmonary Embolism Groups.

Parameter	PE Positive (n = 93)	PE Negative (n = 74)	Difference (95% CI)	*p* *
Heart Rate	102 (84–119)	90 (74–111)	12 (1–17)	0.022
CRP (mg/L)	448 (142–105)	32 (111–139)	416 (219–466)	<0.001
D-dimers (µg/L)	721 (3.74–352)	127 (08–41)	60 (322–656)	<0.001
hs-Troponin (ng/L)	2735 (34–917)	113 (31–187)	158 (74–203)	<0.001
NT-proBNP	641 (171–2745)	177 (59–814)	464 (94–635)	0.002
PaO_2_ (kPa)	832 (728–1000)	1075 (979–1200)	−24 (−296–−170)	<0.001

* Mann–Whitney U test results are presented as: median (IQR). Abbreviations: CRP—C-reactive protein, NT-proBNP—Pro-brain natriuretic peptide, PaO_2_—partial oxygen pressure in blood.

**Table 3 healthcare-13-00011-t003:** ROC (receiver operating characteristic) analysis results: odds ratio, Youden index (J), and cut-off for sensitivity and specificity.

Odds ratio	6.099
OR 95% CI	3.339–11.139
Youden index (J)	0.537
Associated ratio	>1.483
Sensitivity	75.27%
Specificity	78.38%

## Data Availability

The original data presented in this study are openly available in FigShare at DOI: https://doi.org/10.6084/m9.figshare.27117142.v1.

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
