# Peer review of "Diagnostic Value of the Alveolar–Arterial Oxygen Gradient in Pulmonary Embolism: A Cross-Sectional Study"

_healthcare, 2024, doi:10.3390/healthcare13010011_

Round 1

Reviewer 1 Report

Comments and Suggestions for Authors

The authors performed a single-center retrospective analysis of 168 patients with confirmed PE to evaluate the diagnostic value of the Aa gradient. The authors found that incorporating the Aa gradient into the PE workup improves early detection of PE.

Major concerns

The initial premise of this study is that PE is often misdiagnosed due to it’s specific symptoms. I strongly disagree with this statement. PE – as the authors mention- is a very common disease and physicians have a high level of suspicion if a patients present with shortness of breath or chest pain. The references provided by the authors also do NOT justify the statement that many PEs are missed. With the advent of the CT, the diagnostic accuracy for PE has increased.

The 2nd major concern is that the Aa gradient is highly influenced by many factors, such as respiratory drive, atelectasis, PNA, and any other lung pathology. Furthermore, it is almost impossible to predict the correct respiratory quotient for each individual patient (we just assume it is 0.8). The very large confidence interval of the Aa gradient makes it almost impossible to define a cut-off value that is applicable across different patient populations. For instance, the Aa gradient changes with age, hemoglobin levels, and is dependent on the PaO2 (at lower PaO2 levels the Hb dissociation curve narrows the Aa gradient).

The authors define “massive PE” based on imaging. This is flawed: massive PE is defined by the presence of hypoperfusion from right heart failure due to the PE obstruction pulmonary blood flow, not based on the location or mass of the clot.

In summary, the diagnosis of PE is straight forward in most cases and a biomarker such as the Aa gradient that is highly non-specific is not needed. Cardiac biomarkers, imaging findings on CT, and echocardiography provide a solid frame work for risk stratification. I do not believe that incorporation of the Aa gradient will add to the diagnostic workup nor risk stratification for acute PE.

Comments on the Quality of English Language

see above

Author Response

Thank you very much for the time to review this manuscript. Your comments and suggestions are appreciated and helped improve the quality of the manuscript. 

Comment: The initial premise of this study is that PE is often misdiagnosed due to its specific symptoms. I strongly disagree with this statement. PE – as the authors mention – is a very common disease, and physicians have a high level of suspicion if a patient presents with shortness of breath or chest pain. The references provided by the authors also do NOT justify the statement that many PEs are missed. With the advent of the CT, the diagnostic accuracy for PE has increased.

Response: Our premise was based on observations that CT pulmonary angiography (CTPA) is frequently utilized, often resulting in a significant proportion of negative findings, which can be costly and resource-intensive. We have many concerns from radiology department stating there are too many negative PE angio sent from emergency department. Many emergency department physicians rely heavily on D-dimer testing which are only good to rule out PE leading to unnecessary imaging in cases where PE is unlikely. We have removed implications suggesting widespread misdiagnosis from the manuscript .We have revised the introduction to reflect that while PE diagnosis has improved significantly, there are still times where diagnosis is delayed due to the non-specificity of symptoms in certain populations. 

Comment: The A-a gradient is highly influenced by many factors, such as respiratory drive, atelectasis, PNA, and other lung pathologies. It is almost impossible to predict the correct respiratory quotient for each individual patient (we just assume it is 0.8). The very large confidence interval of the A-a gradient makes it almost impossible to define a cut-off value that is applicable across different patient populations. For instance, the A-a gradient changes with age, hemoglobin levels, and is dependent on PaO2 (at lower PaO2 levels, the Hb dissociation curve narrows the A-a gradient).

Response: We acknowledge the inherent variability of the A-a gradient and have expanded the limitations section, including its dependency on age, respiratory quotient, hemoglobin levels, and other lung pathologies. We emphasize that the A-a gradient should not be used as a standalone diagnostic tool but as part of a comprehensive diagnostic framework.

Comment 3: The authors define “massive PE” based on imaging. This is flawed: massive PE is defined by the presence of hypoperfusion from right heart failure due to PE obstructing pulmonary blood flow, not based on the location or mass of the clot.

Response: We have corrected the definition of "massive PE" in the methods section. It now aligns with clinical guidelines, focusing on hemodynamic instability and right heart failure instead of clot size or location.

Comment 4: In summary, the diagnosis of PE is straightforward in most cases, and a biomarker such as the A-a gradient, which is highly non-specific, is not needed. Cardiac biomarkers, imaging findings on CT, and echocardiography provide a solid framework for risk stratification. I do not believe that incorporation of the A-a gradient will add to the diagnostic workup nor risk stratification for acute PE.

Response: We have added in the discussion to clarify that the A-a gradient is not proposed as a primary diagnostic tool but rather as an auxilliary marker. Its role is particularly emphasized in scenarios involving low-risk patients where unnecessary imaging can be avoided. This supplementary role is highlighted rather than suggesting it replaces established diagnostic methods.

Other: We have carefully reviewed and improved the language in the manuscript to enhance clarity and readability.

Reviewer 2 Report

Comments and Suggestions for Authors

This analysis prescribes the usefulness of Alveolar-Arterial Oxygen Gradient in patients with pulmonary embolism.

General concept Comments

1. Title: which analysis? Prescribe type of study.

2. Abstract: optimal written.

3. Introduction:
- Line 39: please prescribe also others pre-test probability scores.

- Line 58-59: this sentence ’’[…] Although CTA is currently the gold standard for the diagnosis of PE, this imaging modality is expensive and not as widely available.’’ - is very subjective. Please confirm your opinion with example of price compared to other modalities. You can mention here about pregnant women in which we should try to avoid any radiation. This is good argument for using other way to estimate PE propability.

3. Materials and Methods:
- Line 66: based on which criteria?

- Line 82-84: I am not sure if these circumstances are realistic. Normally the first aid after admission, sometimes also before anamnesis, also in ambulance, is oxygen substitution! This is an first ethical issue and first treatment of any respiratory problem. Maybe some patients received already oxygen but only for arterial blood gas analysis were interrupted. Please analyse the data once again.

- Line 85: Why age 17? Not 18?

- Line 90: prescribe criteria for DVT signs used in these patients.

- Temperature of the patient was also measured? pH? If yes, are you sure that all of these patients has not any significant difference of pH-value?

- Please compare Hb levels and cardiac output in particular groups. It was any difference?

4. Statistical analysis: appropriately prescribed and used.

5. Results:

- which part of patients has additional changes in CT scan? For example, cardiac decompensated state or pleural effusion? Atelectasis? I cannot imagine that every PE patient has clear lungs and only pulmonary embolism lesions.

- Could you show the distribution of A-a gradient divided into the groups conserned burden of thrombus material? For example: subsegmental vs. massive embolism.

6. Discussion:

- Please comment if you recommend the A-a gradient also for patient with segmental or subsegmental pulmonary embolism? Why?

7. Conclusion:

- Line: 259: which ’’clinical scores’’?

Comments on the Quality of English Language

Please adjust your English fluency. 

Author Response

Comment 1: Title: Which analysis? Prescribe type of study.

Response: The title will be updated to specify that this is a "Cross-Sectional study."

Comment 2: Abstract: Optimally written.

Response: Thank you. No changes were made to the abstract.

Comment 3: Introduction: Line 39: Please prescribe also other pre-test probability scores.

Response: We added other pre-test probability scores such as the Geneva score and YEARS criteria. 

Comment 4: Line 58-59: This sentence, “[…] Although CTA is currently the gold standard for the diagnosis of PE, this imaging modality is expensive and not as widely available,” is very subjective. Please confirm your opinion with an example of price compared to other modalities. You can mention here about pregnant women in whom we should try to avoid any radiation. This is a good argument for using other ways to estimate PE probability.

Response: We revised this sentence to compare the cost and accessibility of CT with other modalities and highlighted its limitations for specific populations, such as pregnant women. This adds to the argument for adjunctive diagnostic tools like the A-a gradient.

Comment 5: Materials and Methods: Line 66: Based on which criteria?

Response: We clarified that inclusion criteria were based on clinical suspicion of PE, Wells criteria and arterial blood gas analysis performed upon admission.

Comment 6: Line 82-84: I am not sure if these circumstances are realistic. Normally the first aid after admission, sometimes even before anamnesis, also in ambulances, is oxygen substitution! This is an ethical issue and the first treatment for any respiratory problem. Maybe some patients received oxygen, but it was interrupted for arterial blood gas analysis. Please analyze the data once again.

Response: We comfirm that arterial blood gas analyses were performed before oxygen administration to maintain consistency. This protocol has been detailed in the methods section. We did not include patients whose only ABG were with additional oxygen. 

Comment 7: Line 85: Why age 17? Not 18?

Response: The inclusion of patients aged 17 was based on hospital policy, which allows emergency admissions for younger individuals under certain conditions. This has been clarified in the methods section.

Comment 8: Please compare Hb levels and cardiac output in particular groups. Was there any difference?

Response: We did not collect data on Hb levels or cardiac output in this study. This limitation has been added in the revised manuscript to ensure transparency.

Comment 9:
Which part of patients has additional changes in CT scan? For example, cardiac decompensated state or pleural effusion? Atelectasis? I cannot imagine that every PE patient has clear lungs and only pulmonary embolism lesions.

Response:
We have added information about additional CT scan findings, including pleural effusions, atelectasis, and other associated findings in patients with PE.

Comment: Could you show the distribution of A-a gradient divided into the groups concerned with the burden of thrombus material? For example: subsegmental vs. massive embolism.

Response: Thank you for your suggestion. We have now included an analysis of the A-a gradient across thrombus burden groups (no embolism, massive, lobar, segmental, and subsegmental embolism). Median A-a gradient values and interquartile ranges were calculated for each group, and the results are presented in Figure 4 and described in the Results section.

Our analysis revealed that the A-a gradient values in the massive, lobar, segmental, and subsegmental embolism groups were significantly higher than in the no-embolism group (P < 0,000001). However, no significant differences were observed between the embolism groups (massive, lobar, segmental, and subsegmental). These findings are now discussed in the manuscript, highlighting the sensitivity of the A-a gradient to gas exchange abnormalities across all thrombus burden groups.

Additionally, we have added in the Discussion section that the assessment of thrombus burden was performed by a single radiologist, which leads to potential subjectivity. 

Comment:
Please comment if you recommend the A-a gradient also for patients with segmental or subsegmental pulmonary embolism? Why?

Response:
Thank you for this comment. We have expanded the discussion to add the potential utility of the A-a gradient in patients with segmental and subsegmental PE. While the sensitivity of the A-a gradient may decrease in cases of lower thrombus burden, it can still give a valuable diagnostic information when used in combination with clinical scores and D-dimer testing. This addition can be found in the Discussion section. 

Comment:
Line 259: Which "clinical scores"?

Response: We have clarified that the clinical scores referenced are the Wells criteria and Geneva score. This clarification has been added to the Conclusion

Comment: Please adjust your English fluency.

Response: We have carefully reviewed and improved the language throughout the manuscript to enhance clarity and readability. 

Reviewer 3 Report

Comments and Suggestions for Authors

Thank you for the opportunity to review this paper.

The aim of this retrospective study of 167 patients was to rassess the role of A-a O2 gradient and  of the ratio between the expected and observed A-a O2 gradient, as a diagnostic tool for the early detection of PE, in combination with other markers.

 Minor comments:

- One of the exclusion criteria was a not clearly result of CT , "PE diagnosis as possible" (line 81). The results of this study suggest (despite the limited number of patients) that the A-a ratio is a useful diagnostic tool that may be included in the screening algorithms for PE diagnosis (line 218), as the authors rsuggest , and possible avoiding the CT in low risk patients. Please specify if that is for low probability for PE patients)

- Please reformulate  the phrase "incorrect positive diagnosis"(line 193), maybe replace it with "false positive"

-As the authors have mentioned, the role of the A-a gradient  for therisk stratification of these patients was not a aim of this study.  Add a comment about the line 239, and the corellation with The data from the Table 1, from the PE positive CT findings (massive, lobar, left/right sided etc)

-Add comments about the markers used for the PE diagnosis in clinical practice (D-Dimers, hsTn) and risk stratification(i.e RV-right ventricle acute failure0 and not CRP or NT-proBNP( more useful for differential diagnosis)

-Add more comments about the subset of patients with normal A-a value and confirmed PE(Fig1)

Author Response

Thank you for your review.

Comment 1: One of the exclusion criteria was a not clearly defined result of CT, “PE diagnosis as possible” (line 81). The results of this study suggest (despite the limited number of patients) that the A-a ratio is a useful diagnostic tool that may be included in screening algorithms for PE diagnosis (line 218), possibly avoiding CT in low-risk patients. Please specify if this is for low-probability PE patients.

Response: We have clarified in the discussion that the A-a gradient is recommended for use in low-probability PE patients to minimize unnecessary CT scans.

Comment 2: Please reformulate the phrase “incorrect positive diagnosis” (line 193), maybe replace it with “false positive.”

Response: The phrase has been replaced with “false positive diagnosis”.

Comment 3: Add comments about the markers used for PE diagnosis in clinical practice (D-dimers, hs-Troponin) and risk stratification (e.g., RV-right ventricle acute failure) and not CRP or NT-proBNP (more useful for differential diagnosis).

Response: We expanded the discussion to include the roles of D-dimers and hs-Troponin in diagnosis and RV strain assessment in risk stratification. CRP and NT-proBNP were discussed in the context of differential diagnosis.

Comment :
As the authors have mentioned, the role of the A-a gradient for the risk stratification of these patients was not an aim of this study. Add a comment about line 239 and the correlation with the data from Table 1, from the PE positive CT findings (massive, lobar, left/right-sided, etc.).

Response:
Thank you for this suggestion. We have now included an analysis of the A-a gradient across thrombus burden groups (no embolism, massive, lobar, segmental, and subsegmental embolism). Median A-a gradient values and interquartile ranges were calculated for each group, and the results are presented in Figure 4 and described in the Results section. Our analysis revealed that the A-a gradient values in the massive, lobar, segmental, and subsegmental embolism groups were significantly higher than in the no-embolism group. However, no significant differences were observed between the embolism groups (massive, lobar, segmental, and subsegmental). These findings are now discussed in the manuscript, highlighting the sensitivity of the A-a gradient to gas exchange abnormalities across all thrombus burden groups.

Comment:
Add more comments about the subset of patients with normal A-a values and confirmed PE (Fig. 1).

Response:
Thank you for highlighting this important subset. We have expanded the Discussion section to address the physiological mechanisms that might explain normal A-a values in patients with confirmed PE. These patients may represent cases where compensatory mechanisms maintain normal arterial oxygen levels despite ventilation-perfusion mismatch. 

Round 2

Reviewer 2 Report

Comments and Suggestions for Authors

Dear Authors,

Thank you for the prompt submission of the revised manuscript. I appreciate the effort you have made to address my comments and make the necessary corrections.

Thank you for your valuable contribution. 

Author Response

Dear Reviewer,

Thank you for taking the time to review our revised manuscript and for your kind words. We greatly appreciate your constructive feedback, which has helped us improve the quality and clarity of our work.